# Potential Azo-8-hydroxyquinoline derivatives as multi-target lead candidates for Alzheimer's disease: An in-depth in silico study of monoamine oxidase and cholinesterase inhibitors

**Fatima Zahra Guerguer[1], Bouchra Rossafi[1], Oussama Abchir[1], Yasir S. Raouf[2], Dhabya Bakhit Albalushi[2], Abdelouahid Samadi[2]\*, Samir Chtita[1]\***

1 Laboratory of Analytical and Molecular Chemistry, Faculty of Sciences Ben M'Sik, Hassan II University of Casablanca, Casablanca, Morocco, 2 Department of Chemistry, College of Science, United Arab Emirates University, Al Ain, United Arab Emirates

\* samadi@uaeu.ac.ae (AS); samirchtita@gmail.com (SC)

## Abstract

Cognitive dysfunction in Alzheimer's disease results from a complex interplay of various pathological processes, including the dysregulation of key enzymes such as acetylcholinesterase (AChE), butyrylcholinesterase (BuChE), and monoamine oxidase B (MAO-B). This study proposes and designs a series of novel molecules derived from 8-hydroxyquinoline (Azo-8HQ) as potential multi-target lead candidates for treating AD. An exhaustive *in silico* analysis was conducted, encompassing docking studies, ADMET analysis, density functional theory (DFT) studies, molecular dynamics simulations, and subsequent MM-GBSA calculations to examine the pharmacological potential of these molecules with the specific targets of interest. Out of the 63 Azo-8HQ derivatives analysed, two molecules, 14c and 17c, demonstrated strong affinities for AChE, BuChE, and MAO-B, along with favourable pharmacokinetic profiles and electronic properties. Molecular dynamics simulations confirmed the stability of these molecules within the active sites of the targets, and MM-GBSA calculations revealed low binding energies, indicating robust interactions. These findings identify molecules 14c and 17c as promising multi-target candidates for the treatment of AD, based on an in-depth computational study aimed at minimizing drug development costs and time. Future work will include the synthesis of these molecules followed by in-depth *in vitro* and *in vivo* testing to validate their potential therapeutic efficacy.

## Introduction

Cognitive dysfunctions in Alzheimer's disease (AD) encompass challenges with learning, memory, processing speed and persistent deficits in mental flexibility, attention, and behaviour [1]. Accounting for approximately 60% of all dementia cases, AD has been recognized by the World Health Organization (WHO) as a global public health priority due to the lack of a

**Data Availability Statement:** All relevant data are within the paper and its Supporting Information files.

**Funding:** AS and YSR acknowledge the support of the UAEU through an internal Start-up grant 2023 (Grant Code G00004400) and an internal Start-up grant 2024 (Grant 12S156), respectively.

**Competing interests:** The authors have declared that no competing interests exist.

definitive curative treatment [2]. The disease is characterized by several major neuropathological abnormalities: the loss of cholinergic neurons, excessive breakdown of acetylcholine by acetylcholinesterase (AChE), oxidative stress in neurons, and the development of amyloid plaques and neurofibrillary tangles [3]. Despite numerous attempts to target these pathological features, current therapeutic approaches have often fallen short due to an incomplete understanding of the underlying mechanisms of the disease [4]. Existing treatments primarily target AChE inhibition, with drugs such as donepezil, rivastigmine, and galantamine. Memantine addresses glutamate-induced excitotoxicity. However, these therapies mainly provide symptomatic relief without tackling the root causes of the disease [5].

Among the various hypotheses proposed to explain the pathogenesis of AD, the cholinergic hypothesis is particularly prominent. This hypothesis suggests that the decline in cholinergic activity in the brain, particularly in the cerebral cortex and hippocampus, plays a key role in the cognitive deterioration observed in AD patients [6]. Acetylcholine (ACh) found in postsynaptic neurons and neuromuscular regions, is essential for nerve impulse transmission. The degradation of ACh by AChE leads to significant cognitive impairments and loss of muscle coordination. Inhibition of AChE slows the breakdown of ACh in the brain, thus providing symptomatic relief to patients [7, 8]. Additionally, BuChE also plays a role in regulating ACh levels. Although its role is less central than that of AChE, BuChE becomes more active as the disease progresses, partially compensating for the reduced activity of AChE. Besides inhibiting AChE, BuChE inhibition can provide additional benefits by maintaining higher ACh levels, thus improving cognitive and motor functions in AD patients [9]. Furthermore, amine homeostasis in the brain is disrupted by the oxidative deamination of monoamine oxidase (MAO), an enzyme involved in the catabolism of neurotransmitter monoamines [10]. MAO-B, primarily located in glial cells, increases its activity with age, contributing to oxidative stress and neurodegeneration. In the context of AD, an increase in MAO-B activity has been observed, which can amplify neuronal oxidative stress and promote the aggregation of pathological proteins such as amyloid plaques. Therefore, MAO-B inhibitors are considered a potential therapeutic strategy to reduce oxidative stress and slow the progression of AD [11, 12].

Given the complexity of AD and the multitude of factors contributing to cognitive decline, it is imperative to develop multi-target directed ligands (MTDLs) for effective disease management. The conventional "one target-one molecule" approach is inadequate for addressing the multifaceted pathological mechanisms of AD, as this method fails to address the multifactorial nature of the disease [13, 14].

Computer-aided drug design (CADD) represents a significant advancement in the development of effective therapies for complex diseases such as AD [15]. CADD utilizes computational techniques to identify, design, and optimize new therapeutic molecules, thereby reducing the time and costs associated with drug development. Through in silico simulations, it is possible to predict the interaction of molecules with their biological targets, evaluate their pharmacokinetic and toxicity properties, and optimize their affinity and selectivity [16, 17].

Hydroxyquinoline derivatives are recognized as promising candidates in the fight against AD. These compounds have demonstrated significant inhibitory activity against various enzymes involved in AD pathogenesis. A multi-target compound has recently been highlighted for its remarkable efficacy in inhibiting amyloid-β (Aβ) aggregation and its antioxidant properties [18, 19]. Additionally, hydroxyquinoline derivatives show selective inhibition of BuChE and MAO-B and the ability to chelate metals in the brain, thereby enhancing their potential as treatments for AD [20].

This current study focuses on the design of new 8-hydroxyquinoline-derived molecules, such as phenylazo-8-hydroxyquinoline (Azo-8HQ), for evaluation as multitarget drugs against AD. A series of these derivatives was proposed and subjected to a comprehensive in silico

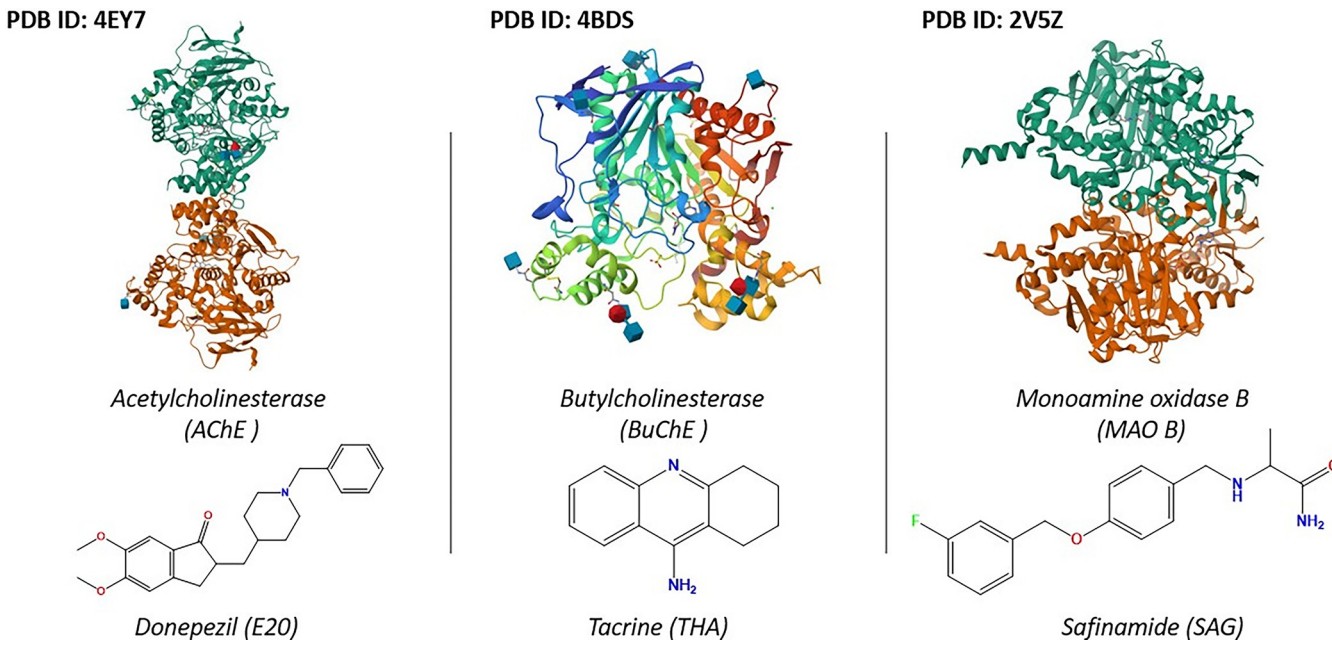

**Fig 1. PDB crystallographic structures and reference ligands of the selected target protein complexes.**

study to assess their activity against three key enzymes implicated in AD pathology: AChE, BuChE, and MAO-B (Fig 1). We applied a range of in silico analyses, including docking studies to evaluate the binding affinity of the compounds and ADMET profiling to estimate their pharmacokinetic properties and safety. Additionally, DFT studies were conducted to gain deeper insights into the electronic properties of the molecules, and molecular dynamics simulations followed by MM-GBSA calculations, were performed to rigorously assess the binding interactions and stability of these molecules within the enzyme active sites.

## Material and methods

### Data sources

A series of novel molecules derived from Azo-8-hydroxyquinoline, consisting of 63 derivatives, has been proposed and designed. These structures were optimized using the MMFF94 force field and the steepest descent algorithm through the Avogadro program, and subsequently saved in PDB format. The 2D structures of the studied molecules are listed in S1 Table.

### Molecular docking analysis

Molecular docking is a method used to predict the optimal binding position of a ligand with its target enzyme, providing insights into potential intermolecular interactions, free energies of binding and the stability of the formed binary complex [21]. In this study, the series of Azo-8HQ derivatives were docked against the three enzymes, namely AChE, BuChE, and MAO-B using the Schrödinger Release 2020–3: Maestro software suite, following the structured protocol outlined below:

### Protein preparation

The crystallographic structures of the three proteins AChE, BuChE, and MAO-B along with their co-crystallized small molecule inhibitors (ligands), were obtained from the Protein Data

Bank ([www.rcsb.org](www.rcsb.org)) using the PDB identifiers 4EY7, 4BDS and 2V5Z, respectively. Before conducting the analysis, the proteins were prepared using the Protein Preparation Wizard tool. Issues such as atom overlaps, alternate positions, missing atoms, and incorrect atom types were identified and corrected by adding hydrogen atoms, adjusting atom positions, minimizing the structures, and fixing missing side chains. Additionally, water molecules located beyond 0.5 Å were removed, and the proteins were further processed in preparation for the study.

## Ligand preparation

The previously optimized ligands were further prepared using the LigPrep tool under the OPLS3e force field. The library underwent energy minimization. The software was configured to generate a maximum of 32 conformers for each ligand, with states generated at a pH of $7 \pm 2$.

## Molecular docking

The receptor grids for the three proteins were generated using Maestro's Receptor Grid Generation tool, based on the co-crystallized ligands of the protein structures. The series of compounds underwent docking against AChE, BuChE and MAO-B using two different Glide docking modes: SP and XP. These modes vary in precision, speed, and scoring function. Glide SP strikes a balance between speed and precision, typically requiring around 10 seconds per compound for exhaustive sampling [22]. XP mode focuses on eliminating false positives and penalizes molecules with low binding affinity to the receptor [23].

For each target protein, the docking protocol utilized the same approach with a specific standard ligand, utilizing Glide score as the criterion to evaluate library compounds. Donepezil (DN) served as the standard ligand for AChE (PDB ID: 4EY7) due to its potent inhibitory activity against this enzyme [24]. Respective co-crystallized ligands were employed as standards for BuChE (PDB ID: 4BDS) and MAO-B (PDB ID: 2V5Z).

## Drug likeness and ADMET prediction

The evaluation of pharmacokinetic properties plays a crucial role in the therapeutic validation of our selected molecules. These characteristics encompass essential aspects such as absorption, distribution, metabolism, excretion, and toxicity (ADMET) [25]. Using canonical SMILES, we utilized the Swiss ADME web server ([http://swissadme.ch/](http://swissadme.ch/)) for a thorough analysis of drug-likeness, assessing adherence to the rule of five for drug similarity. Additionally, the pkCSM server ([https://biosig.lab.uq.edu.au/pkcsm/](https://biosig.lab.uq.edu.au/pkcsm/)) was employed for a detailed evaluation of ADMET profiles, providing crucial insights to predict and understand the in vivo behaviour of our molecules [26]. This integrated approach enabled a rigorous computational assessment, facilitating the identification of compounds that meet both therapeutic potential and bioavailability requirements, while strictly adhering to safety standards [27].

## Molecular quantum analysis

Density Functional Theory (DFT), a quantum mechanical approach, examines the electronic density $\rho(r)$ in relation to the wave function $\Psi(r_1\sigma_1, r_2\sigma_2, \ldots r_n\sigma_n)$, where r and σ denote the spatial and spin coordinates of all electrons in the atom or molecule [28]. Data on the electronic state of a molecule are crucial for assessing its reactivity by identifying sites for attack or addition, elucidating the nature of the studied reagent. Global reactivity descriptors [29], such as band gap energy, electronegativity, molecular hardness, molecular softness and HOMO/

## Ionization potential

$$IP = -EHOMO$$

## Electron affinity

$$EA = -ELUMO$$

## Energy gap

$$Egap = ELUMO - EHOMO$$

## Hardness

$$\eta = \frac{IP - EA}{2}$$

## Softness

$$\sigma = \frac{1}{\eta}$$

## Electronegativity

$$\chi = \frac{IP + EA}{2}$$

## Chemical potential

$$\mu = -\chi$$

## Electrophilicity index

$$\omega = \frac{\mu^2}{2\eta}$$

**Fig 2. Calculation formulas for the main molecular descriptors used in the DFT analysis.**

LUMO orbital energies, are commonly used in DFT to evaluate chemical reactivity. This approach aims to understand the stability of protein-ligand complexes without inducing unwanted chemical reactions that could yield harmful byproducts to the human body [30]. In our study, an extensive quantum optimization via DFT was performed on the selected molecules, previously optimized using the MMFF94 force field and steepest descent algorithm in the Avogadro program. This optimization was conducted using Gaussian [31], employing the DFT B3LYP method [32] and the high-level 6-311G (d, p) quantum database [33] to obtain electronic state spectra, HOMO and LUMO orbital energies ($E_{HOMO}$, $E_{LUMO}$). These data facilitated the calculation of relevant molecular descriptors (Fig 2).

### Molecular dynamics simulation

Molecular dynamics simulation plays a pivotal role in our study, elucidating intricate molecular interactions [34]. We employed Desmond, a cornerstone of the Schrödinger Release 2020–3: Maestro software suite, to investigate how our selected molecules interact with the three targets studied. Each simulation spanned 200 nanoseconds, meticulously prepared using Schrödinger's Protein Preparation Wizard to maintain a consistent pH of 7.4. The System Builder facilitated the creation of a realistic simulation environment, utilizing the TIP3P water model for solvation in a orthorhombic simulation box (10 Å × 15 Å × 20 Å) [35]. To ensure electrical neutrality, Na+/Cl− counter-ions were added at a concentration of 0.15 M. The robust OPLS3e force field was instrumental in configuring our systems [36]. Following initial equilibration, simulations were conducted under various thermodynamic ensembles (NVT and NPT), employing tailored thermostats and barostats to stabilize temperature and pressure. MD simulations were subsequently executed over 100 nanoseconds. Throughout these simulations, meticulous scrutiny of stability parameters such as RMSD, RMSF and protein-ligand

interactions was conducted. These comprehensive investigations shed light on binding affinities and the efficacy of our selected molecules against the three biological targets studied.

## MM-GBSA calculations

To determine the free binding energies, we conducted Molecular mechanics with generalised Born and surface area solvation (MM-GBSA) calculations through Schrödinger's Maestro interface. The MM-GBSA panel within the prime module facilitated the estimation of free binding energies for our selected molecules, with the enzyme included in the workspace and compound selections guided by project table data [37]. The (VSGB) solvation model and OPLS3e force field were employed to enhance the accuracy and reliability of our assessments [38]. These integrated approaches provided a comprehensive understanding of the binding affinities and efficacy profiles of our selected molecules across the three studied targets.

## Results and discussion

### Molecular docking

Before moving on to the main docking experiments, it was essential to conduct re-docking simulations for ligands co-crystallized with various proteins. The tested ligands included donepezil (E20) with AChE (PDB ID: 4EY7), tacrine (THA) with BuChE (PDB ID: 4BDS), and SAG with MAO-B (PDB ID: 2V5Z). The results showed promising accuracy, with RMSD values of 0.38 Å, 0.06 Å, and 0.67 Å respectively (Fig 3). These findings confirm the reliability of the docking protocol used in our study.

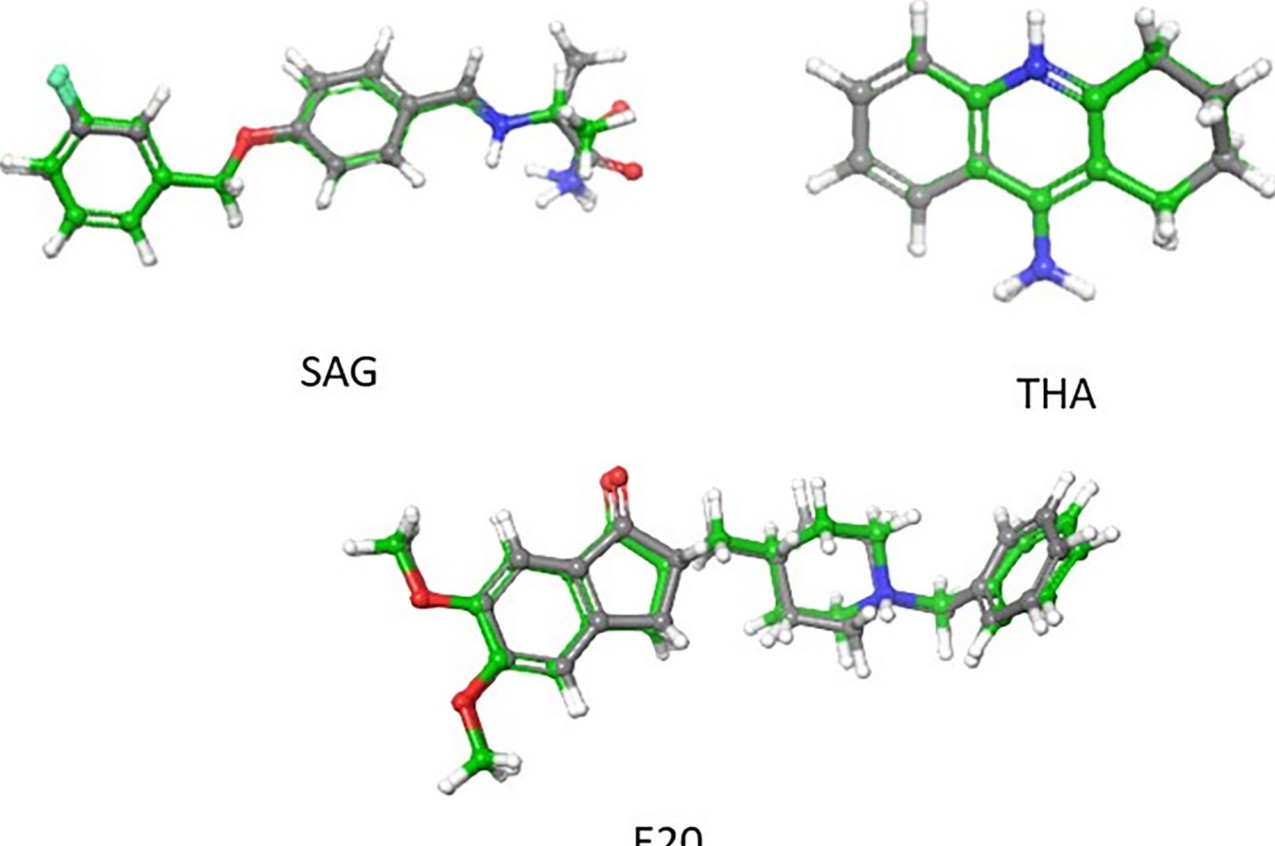

**Fig 3. Superimposed poses of the original (gray) and redocked (green) ligands within the protein receptor pockets.**

**Table 1. Binding affinity of the best complexes formed between docked ligands and target proteins (AChE, BuChE, and MAO-B).**

| Molecule | Docking Scores kcal/mol | | |
|---|---|---|---|
| | AChE | BuChE | MAO-B |
| Co-crystallized ligand | -14.64 | -8.96 | -9.76 |
| **2b** | -14.60 | -9.81 | -9.38 |
| **4b** | -15.24 | -9.04 | -9.25 |
| **7b** | -15.54 | -9.13 | -9.11 |
| **14b** | -14.31 | -9.24 | -9.26 |
| **15b** | -14.70 | -9.48 | -9.23 |
| **17b** | -14.33 | -10.14 | -8.79 |
| **4c** | -16.17 | -10.35 | -11.28 |
| **5c** | -16.40 | -10.70 | -10.19 |
| **9c** | -16.38 | -9.84 | -9.81 |
| **14c** | -15.96 | -9.50 | -9.56 |
| **17c** | -15.94 | -10.79 | -9.67 |
| **20c** | -16.66 | -11.01 | -9.74 |

Following the re-docking process, we conducted molecular docking simulations to assess binding affinity and gain detailed insights into the ligand-protein interactions between the studied molecules and the target proteins. The results, compared to the co-crystallized ligands, are presented in Table 1. Negative binding energy scores indicate increased stability of the ligand-protein complexes, with lower scores signifying stronger and more stable interactions. Our study specifically focused on 12 molecules selected from the initial 63 derivatives (~19% filtering), which exhibited the lowest binding energies, with scores comparable to and even surpassing those of the reference ligands for all three targets. These molecules appear promising for evaluation as multi-target anti-Alzheimer agents, given their ability to simultaneously target several proteins involved in the pathology. Subsequently, these 12 selected molecules were subjected to Drug-likeness and ADMET analysis to evaluate their pharmacokinetic properties.

## Drug likeness and safety profile prediction of the selected molecules

Ensuring the safety and efficacy of drug molecules is crucial in the discovery of new treatments. In silico pharmacokinetic profiling has proven to be a valuable method for evaluating these aspects at the early stages of development [39]. In this study, we assessed the drug-likeness of small molecules based on Lipinski's Rule-of-5 (Ro5), which define acceptable limits for four molecular physicochemical properties for orally active compounds, thus ensuring their initial pharmacological viability. These criteria include a molecular weight below 500 Daltons, a number of hydrogen bond donors (HBD) less than 5, a number of hydrogen bond acceptors (HBA) less than 10, and a partition coefficient (logP) less than 5 [40]. The evaluation of the drug-likeness of the selected molecules, as illustrated in Table 2, reveals promising characteristics suggesting their potential for drug development. All the molecules exhibit a molecular weight below the 500 Dalton threshold set by Lipinski's rule. Additionally, lipophilicity, calculated using the partition coefficient between water and n-octanol, revealed values below 5, indicating potentially favourable absorption. Similarly, the number of hydrogen bond donors and acceptors meets Lipinski's criteria (HBD < 5 and HBA < 10). It is important to note that all the selected molecules comply with Lipinski's rule of five without any detected violations.

**Table 2. Assessment of drug likeness for selected molecules based on Lipinski's criteria.**

| Molecule | Lipinski's Criteria | | | |
|---|---|---|---|---|
| | Molecular Weight | Num. H-Bond Acceptors | Num. H-Bond Donors | logP |
| 2b | 361.35 | 7 | 2 | 1.3 |
| 4b | 384.35 | 8 | 2 | 2.7 |
| 7b | 350.80 | 5 | 2 | 2.6 |
| 14b | 340.38 | 5 | 2 | 2.6 |
| 15b | 330.38 | 5 | 2 | 2.4 |
| 17b | 332.36 | 6 | 3 | 1.6 |
| 4c | 398.38 | 8 | 1 | 3.0 |
| 5c | 364.83 | 5 | 1 | 2.9 |
| 9c | 348.37 | 6 | 1 | 2.8 |
| 14c | 354.40 | 5 | 1 | 2.8 |
| 17c | 346.38 | 6 | 2 | 1.9 |
| 20c | 360.41 | 7 | 1 | 2.0 |

The predicted pharmacokinetic properties (absorption, distribution, metabolism, and excretion) and toxicity of the molecules have been analysed to evaluate their potential as drug candidates, with the results summarized in Table 3. Regarding absorption, all the selected molecules exhibit a Human Intestinal Absorption percentage (HIA %) greater than 80%, indicating good intestinal absorption. The prediction of Caco-2 permeability, used to assess oral drug absorption, reveals that most of the selected molecules have a predicted value above 0.9, making them permeable to Caco-2 cells, with the exception of molecules 4b, 7b, 14b, and 15b. Concerning distribution, effective penetration of drugs across the blood-brain barrier (BBB) is crucial for treating central nervous system (CNS) disorders. molecules with a LogBB value below -1 have limited brain diffusion, while those exceeding 0.3 show increased permeability to the BBB. Additionally, compounds with a Log PS greater than -2 can enter the CNS, whereas those with a Log PS below -3 would face difficulties crossing the CNS [26]. All selected molecules demonstrate the ability to cross the BBB and are classified as active in the CNS, meeting the required thresholds. On the metabolic front, inhibition of cytochrome P450, essential for hepatic drug metabolism, was assessed. Most molecules do not inhibit CYP2C19, CYP2C9, CYP2D6, or CYP3A4 enzymes, thus reducing the risk of hepatic dysfunction and undesirable drug interactions, with the exception of molecules 2b, 17b, 4c, 17c, and 20c, which were found to inhibit CYP2D6 and CYP3A4. Regarding excretion, the molecules exhibit low clearance values, indicating prolonged half-lives and potentially sustained therapeutic effects. Safety is a crucial condition for successful drug development. Therefore, toxicity prediction, including mutagenicity (AMES), hepatotoxicity, cardiotoxicity (hERG), and skin sensitization, was conducted. The results reveal that only molecules 17b, 14c, and 17c present a favourable safety profile, showing no alerts in any of these studies, thereby enhancing their potential as safe lead compound. In contrast, other molecules displayed signs of toxicity, primarily in terms of hepatotoxicity.

## Intermolecular interactions between selected molecules and targets

Based on the results of molecular docking analysis and the prediction of drug-likeness and safety profiles, all other molecules will be eliminated, and only the two molecules, 14c and 17c, were selected for further studies due to their multi-target profile and favourable, physicochemical properties, and non-toxic characteristics. Molecule 14c, targeting three enzymes,

Table 3. ADMET profiles of selected molecules.

| Molecule | Absorption | | Distribution | | Metabolism | | | | | Excretion | | | Toxicity | |
| | CaCO2 | HIA % | BBB Permeability (log BB) | CNS Permeability (log PS) | CYP2C19 Inhibitor | CYP2C9 Inhibitor | CYP2D6 Inhibitor | CYP3A4 Inhibitor | Total Clearance | AMES | hERG | Hepatotoxicity | Skin Sensitization |
|---|---|---|---|---|---|---|---|---|---|---|---|---|---|
| 2b | 1.02 | 91.5 | -0.55 | -2.18 | No | No | Yes | Yes | 0.31 | Yes | Yes | Yes | No |
| 4b | 0.72 | 87.7 | -0.64 | -1.84 | No | No | No | No | 0.50 | Yes | No | Yes | No |
| 7b | 0.65 | 88.3 | -0.35 | -1.87 | No | No | No | No | 0.57 | Yes | No | Yes | No |
| 14b | 0.75 | 91.1 | -0.10 | -1.96 | No | No | No | No | 0.69 | Yes | No | Yes | No |
| 15b | 0.77 | 90.0 | -0.16 | -1.91 | No | No | No | No | 0.65 | Yes | No | Yes | No |
| 17b | 1.09 | 87.5 | -0.87 | -2.18 | No | No | Yes | No | 0.65 | No | No | No | No |
| 4c | 0.97 | 87.5 | 0.05 | -1.75 | No | No | Yes | No | 0.36 | Yes | No | Yes | No |
| 5c | 0.96 | 88.3 | 0.04 | -1.78 | No | No | No | No | 0.50 | No | No | Yes | No |
| 9c | 0.99 | 89.8 | -0.08 | -1.94 | No | No | No | No | 0.43 | No | No | Yes | No |
| 14c | 0.96 | 90.9 | 0.03 | -1.87 | No | No | No | No | 0.55 | No | No | No | No |
| 17c | 1.11 | 87.2 | -0.877 | -2.092 | No | No | Yes | No | 0.514 | No | No | No | No |
| 20c | 1.32 | 91.2 | -0.687 | -2.91 | No | No | Yes | Yes | 0.298 | No | No | Yes | No |

**14c**　　　　　　　　　　　　　　　　**17c**

**Fig 4. 2D structures of the selected molecules 14c and 17c.**

demonstrated affinity scores of -15.96, -9.50, and -9.56 kcal/mol compared to the reference compounds, which had scores of -14.64, -8.96 and -9.76 kcal/mol for AChE, BuChE, and MAO-B, respectively. Similarly, molecule 17c also exhibited a multi-target profile with affinity scores of -15.94, -10.79, and -9.67 kcal/mol, surpassing the reference scores of -14.64, -8.96, and -9.76 kcal/mol. These two molecules were further analysed for their interactions with key amino acid residues, using the reference compounds for each enzyme as controls. Fig 4 shows the 2D structures of molecules 14c and 17c.

AChE is a carboxylase hydrolase with a molecular weight of 121.21 kDa, comprising two subunits, each containing 534 amino acid residues. It is well known that the enzyme AChE features two critical binding sites: the catalytic active site (CAS) and the peripheral anionic site (PAS) [41]. The CAS, located within a 20 Å deep gorge, is characterized by the presence of glutamic acid, serine, and histidine residues. Compounds interacting directly with the CAS cause competitive inhibition. The PAS, on the other hand, can influence substrate and inhibitor binding. Interactions at the PAS can inhibit the enzyme by altering the conformation of the catalytic site and inducing steric hindrance [42]. Docking analysis revealed that Donepezil establishes π-cation interactions between its NH+ group and residues Trp88 and Tyr337 in the CAS, as well as a π-π stacking interaction between residue Trp88 and its benzyl aromatic ring. At the PAS, Donepezil induces π-π interactions between the aromatic ring of its dimethoxyindanone and residue Trp286. Molecules 14c and 17c exhibit similar interactions, establishing hydrogen bonds between the hydroxyl group and Arg296, as well as between the nitrogen core of the quinoline and Phe295 in the CAS. Additionally, these molecules form π-π interactions with residue Trp286 through their quinoline benzyl aromatic rings (Fig 5). Although the interactions at the PAS are similar between Donepezil and molecules 14c and 17c, the latter stand out due to their additional hydrogen bonds at the CAS, which may indicate a potentially more effective inhibition of the enzyme.

BuChE is an enzyme with a molecular weight of 85 kDa, composed of four identical subunits, each containing approximately 574 amino acid residues. The active site of BuChE, located in a gorge 20 Å deep, is characterized by a catalytic triad formed by glutamic acid, serine, and histidine residues [43]. In addition to the catalytic triad, the active site includes a choline binding pocket and an acyl binding pocket, while the peripheral anionic site is marked by the presence of Asp70 and Trp82 residues [44]. The main interactions between the reference ligand tacrine and BuChE include π-aromatic stacking with Trp82, facilitating the attraction of tacrine into the deep gorge, as well as a hydrogen bond between the nitrogen core of tacrine and Hip438. Molecule 14c exhibited similar interactions with BuChE, forming π-π stacking

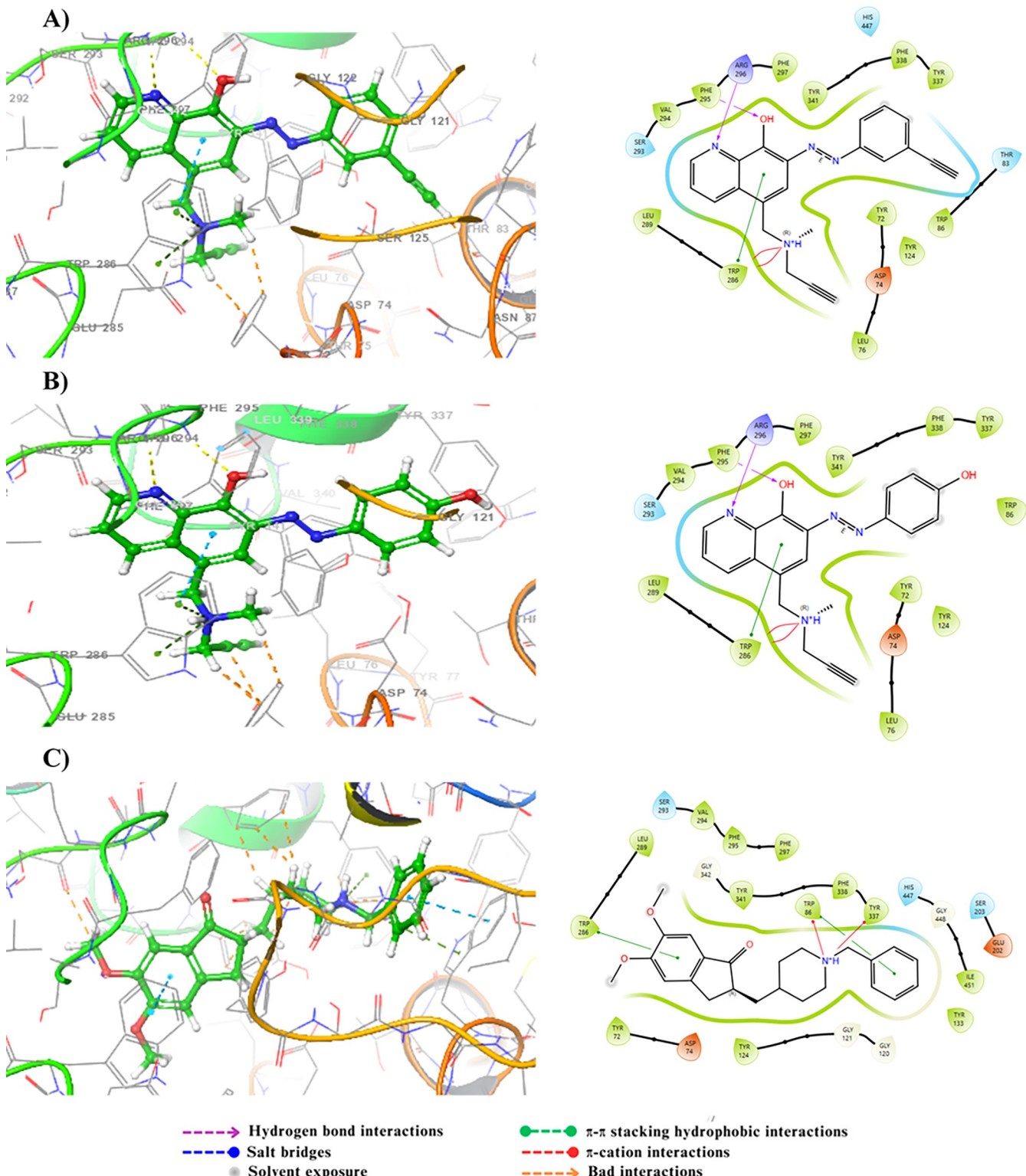

**Fig 5.** 2D and 3D representations of the docking poses for the molecules 14c (A), 17c (B), and the reference (C) in the active site of AChE (PDB ID: 4EY7).

with Trp82 through their aromatic rings. Molecule 17c also showed π-π stacking with Trp82 through their aromatic rings and a hydrogen bond with Thr120 via its hydroxyl group (Fig 6). Although tacrine demonstrates remarkable affinity due to its diverse interactions, molecules 14c and 17c also exhibit promising interactions, such as π-π stacking and hydrogen bonds, suggesting a strong affinity for the active site of BuChE.

For MAO-B, this enzyme is a flavoprotein composed of 520 amino acids, forming two cavities: an entrance cavity and a reactive cavity for substrate binding, delineated by the residues Ile199 and Tyr326. Research has demonstrated that the amino acids Lys296, Trp388, Tyr398, and Tyr435, which form an aromatic sandwich at the substrate binding site, are crucial for MAO-B's catalytic activity [45]. Additionally, the catalytic site of MAO-B includes the residue Ile199, which can adopt either open or closed conformations depending on the bound ligand, thus influencing the flexibility of the catalytic site [46]. Docking results of the reference ligand safinamide revealed that its amine group and ketone group of the amide form two hydrogen bonds with residues Tyr80 and Ser58, respectively. Moreover, π-π interactions were observed between the aromatic ring of safinamide and Tyr228. Molecule 14c exhibits π-π interactions with residues Tyr435 and Tyr228 through its aromatic rings, as well as a hydrogen bond facilitated by its hydroxyl group with Tyr188. Molecule 17c, on the other hand, shows π-π stacking between its aromatic rings and Tyr435, Tyr228, and Tyr298. A hydrogen bond of the hydroxyl group with residue Tyr188 is also observed (Fig 7). These complex interactions highlight the binding affinities of molecules 14c and 17c with MAO-B, emphasizing their significant inhibitory, particularly due to their interactions with essential residues involved in MAO-B inhibition.

## Molecular quantum analysis

Density Functional Theory (DFT) is a Robust and firmly established method in theoretical chemistry, widely used to accurately analyse the electronic properties of molecules by studying their electronic density distribution [47]. Due to its capability to model complex electronic interactions, DFT provides crucial insights into the stability and reactivity of molecules while enabling the optimization and prediction of their structures and molecular orbitals. Frontier molecular orbitals, such as the HOMO (Highest Occupied Molecular Orbital) and the LUMO (Lowest Unoccupied Molecular Orbital), are essential for this assessment. The HOMO reflects the molecule's ability to donate electrons, while the LUMO reveals its capacity to accept electrons [48]. Based on the HOMO and LUMO energies of the two selected molecules, several quantum parameters have been calculated, including: the energy gap Egap, the chemical potential μ, the electronegativity χ, the hardness η, the softness σ, and the electrophilicity index ω. The results are reported in Table 4. The frontier orbitals HOMO and LUMO are also illustrated in Fig 8.

The energy gap Egap is an essential parameter for assessing the stability and reactivity of a molecule. A high Egap value indicates low reactivity, while a lower value suggests increased reactivity. Molecules 14c and 17c exhibit relatively high Egap values of 3.362 and 3.301 eV, respectively, signaling moderate reactivity and enhanced stability. The hardness (η) values for these molecules are also significant: 1.681 eV for 14c and 1.650 eV for 17c. These values indicate a strong ability to maintain their structure under deformation conditions, highlighting remarkable electronic robustness. The low softness (σ) values, 0.594 and 0.605 eV$^{-1}$ respectively confirmed this stability by showing that these molecules are less likely to deform easily and therefore less reactive to perturbations. Regarding electronegativity, which measures an atom's ability to attract electrons, reveals values of 4.347 and 4.101 for molecules 14c and 17c, respectively. These values suggest an increased tendency to attract electrons. In terms of the

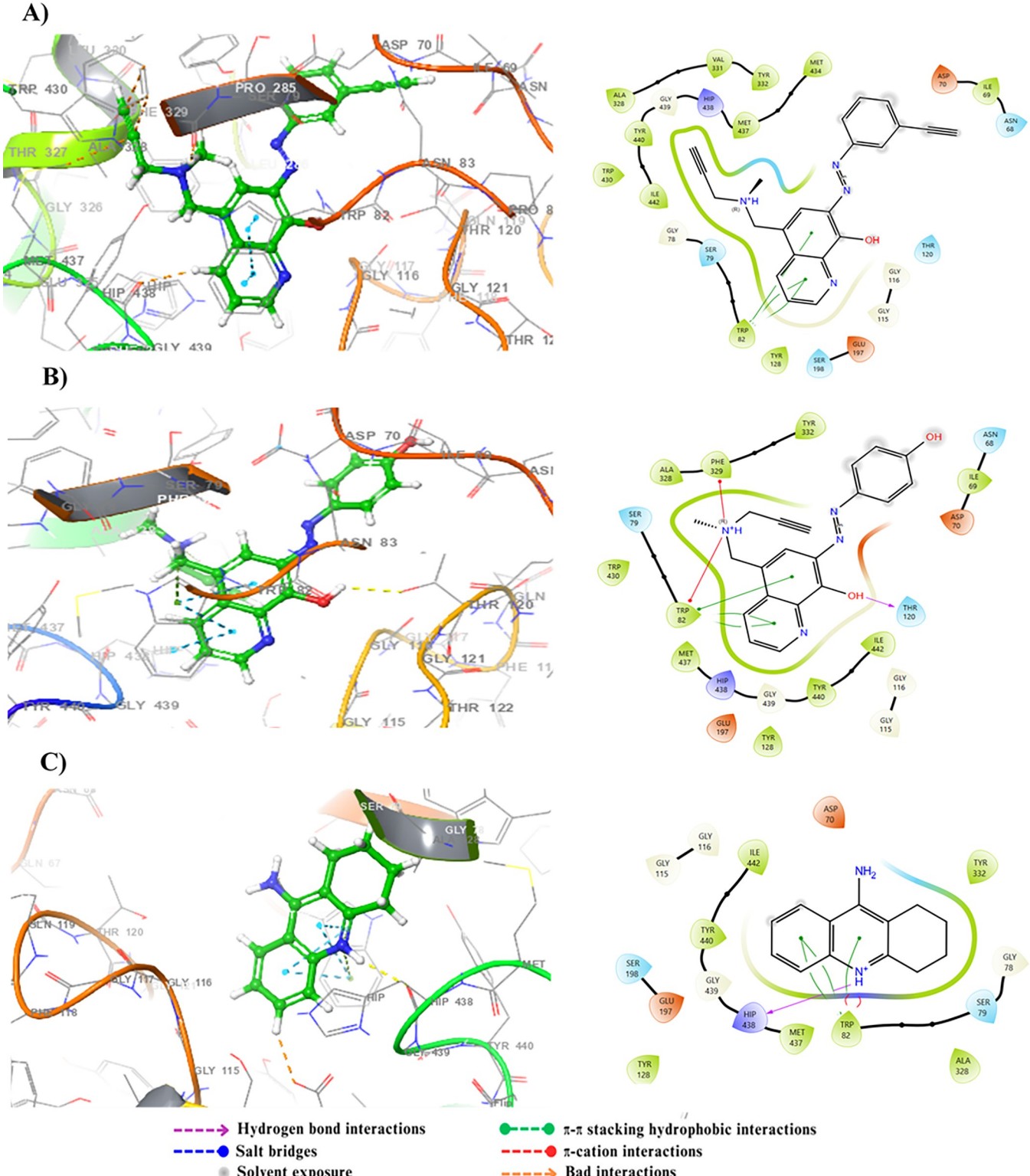

**Fig 6.** 2D and 3D representations of the docking poses for the molecules 14c (A), 17c (B), and the reference (C) in the active site of BuChE (PDB ID: 4BDS).

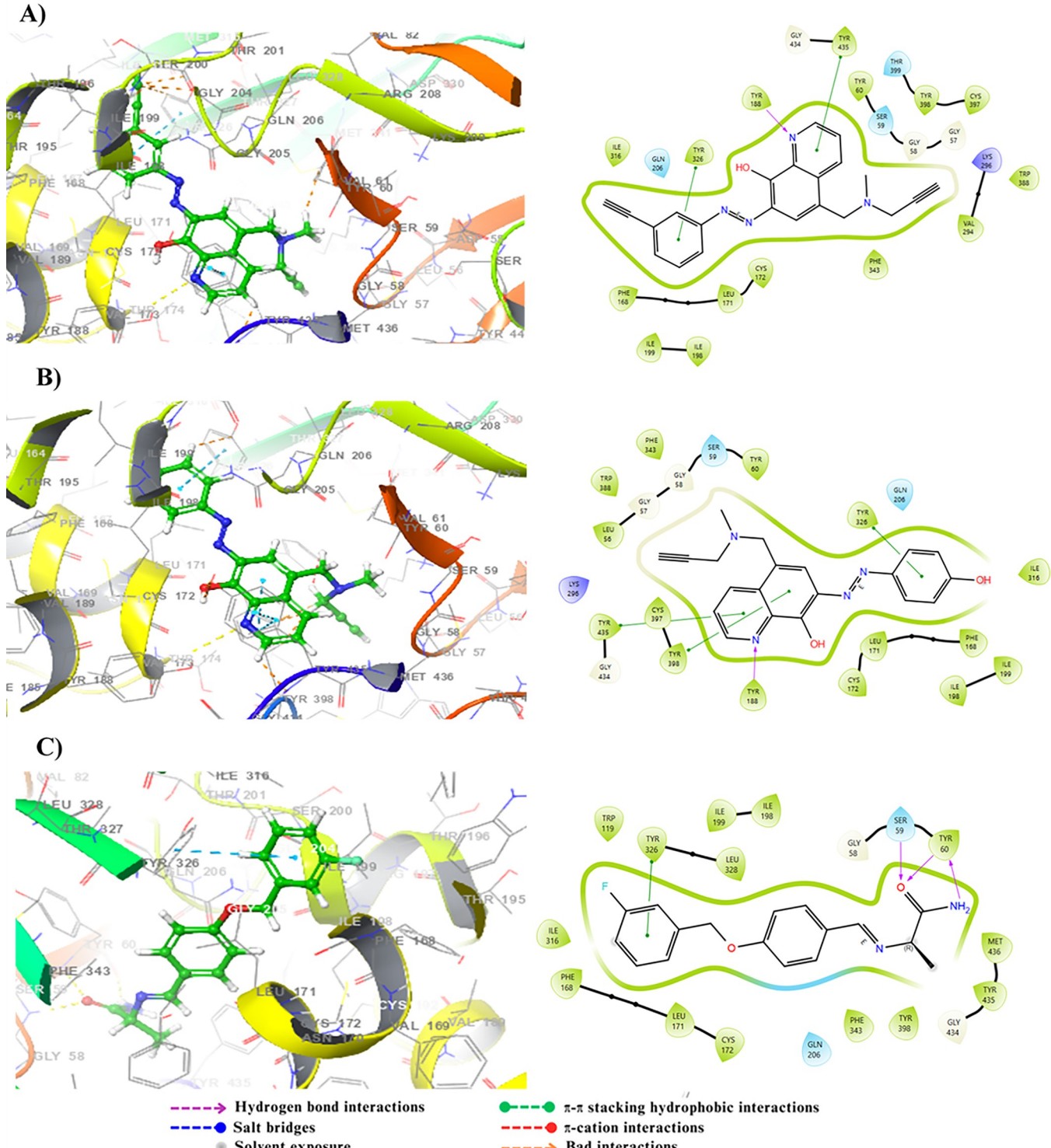

**Fig 7.** 2D and 3D representations of the docking poses for the molecules 14c (A), 17c (B), and the reference (C) in the active site of MAO-B (PDB ID: 2V5Z).

**Table 4. Quantum parameters for selected molecules 14c and 17c.**

| Molecule | Quantum Parameters | |
|---|---|---|
| | **14c** | **17c** |
| $E_{HOMO}$ (eV) | -6.029 | -5.751 |
| $E_{LUMO}$ (eV) | -2.666 | -2.450 |
| Egap (eV) | 3.362 | 3.301 |
| Electronegativity X (eV) | 4.347 | 4.101 |
| Hardness η (eV) | 1.681 | 1.650 |
| Softness σ ($eV^{-1}$) | 0.594 | 0.605 |
| Chemical potential μ (eV) | -4.347 | -4.101 |
| Electrophilicity ω | 5.621 | 5.093 |
| Dipole moment (DEBYE) | 4.340 | 2.839 |

electrophilicity index (ω), which indicates a molecule's tendency to accept electrons in chemical reactions, the obtained values are 5.621 for molecule 14c and 5.093 for molecule 17c. These indices indicate that both molecules are relatively capable of acting as electron acceptors. Finally, the dipole moment, which reflects polarity and the asymmetry in the distribution of electrical charges, is notably higher for molecule 14c at 4.340 Debye compared to 2.839 Debye for molecule 17c.

## Molecular dynamics simulations

To corroborate and deepen our previous findings, we undertook a molecular dynamics study. This computational simulation method allows us to examine the behaviour and stability of ligand-protein complexes under simulated physiological conditions by observing atomic and molecular movements [49]. One of the primary metrics used in this analysis is the Root Mean Square Deviation (RMSD), which evaluates the average deviation of atomic positions relative to a reference structure over time. A low and stable RMSD indicates a stable conformation of the protein-ligand complex. Additionally, the Root Mean Square Fluctuation (RMSF) quantifies the average fluctuation of individual protein residues relative to their mean positions. Higher RMSF values denote more flexible regions, while lower values indicate more rigid areas. Protein-ligand interactions can involve hydrogen bonds, hydrophobic interactions, salt bridges, and other non-covalent forces, all of which are crucial for stabilizing the complex and ensuring the ligand's efficacy as an inhibitor [50]. In our study, we performed molecular dynamics simulations on the two selected molecules interacting with the three targets, AChE, BuChE, and MAO-B. We assessed various parameters to gain insights into their behaviour, stability, and potential as promising inhibitors.

For the AChE target, the RMSD analysis reveals the stability and conformational changes of the protein-ligand complexes over time. The RMSD of the free protein shows that the system reaches equilibrium after approximately 10 ns of simulation, with values stabilizing around 1.2 Å until the end of the simulation. This indicates that the protein maintains a relatively stable conformation without external perturbation. Upon interaction with ligand 14c, the protein exhibits similar stability, with RMSD values fluctuating around 1.4 Å, suggesting a slight structural perturbation compared to the free protein, but overall stability of the complex. However, the complex with ligand 17c shows more significant fluctuations, with RMSD reaching up to 2.5 Å, followed by stabilization around 2.0 Å after 100 ns. This suggests that ligand 17c induces more pronounced conformational changes in the protein compared to 14c, although the complex remains stable. These changes, while more noticeable, fall within an acceptable range, likely due to the conformational adjustment required to accommodate this ligand in the AChE

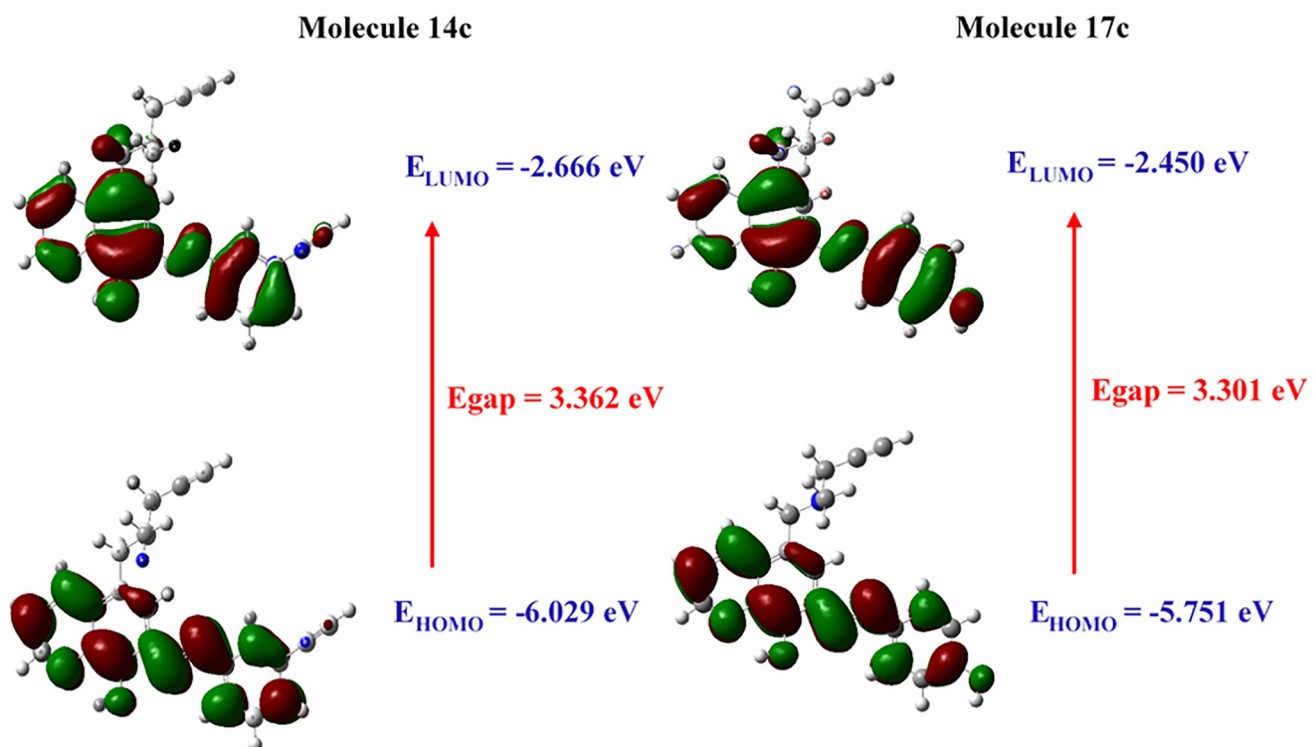

**Fig 8. Molecular orbital distribution plots of HOMO and LUMO of the selected molecules.**

active site (Fig 9). Regarding the RMSD of the ligands, the results show that ligand 14c exhibits greater fluctuations than ligand 17c throughout the simulation. Ligand 14c reaches RMSD values as high as 1.5 Å, particularly after 100 ns, suggesting increased mobility and potential instability within the AChE binding site. However, ligand 17c remains more stable, with lower RMSD values around 1.0 Å, indicating better binding affinity and a more stable interaction with the protein. This increased stability could suggest that ligand 17c has a better fit within the active site of the enzyme, which could be advantageous for its inhibitory activity (Fig 9).

The RMSF analysis shows that the free protein exhibits moderate fluctuations, with RMSF values below 2 Å for the majority of residues such as Gly26, Pro49, and Ile141, indicating an overall stable structure. In the presence of ligand 14c, the fluctuations remain similar, with RMSF values also below 2 Å, suggesting stability comparable to that of the free protein. However, the protein in complex with ligand 17c exhibits more pronounced fluctuations, particularly at residues Ala377 and Pro429, where RMSF values reach 5 Å and 4 Å, respectively, indicating potentially more flexible or exposed regions (Fig 9).

The protein-ligand interactions analysis revealed several significant hydrogen bonds and hydrophobic interactions for both molecules. For molecule 14c, residues Trp286, Arg296, and Phe295 exhibited the strongest interactions, with hydrogen bonds between Phe295 and Arg296 having interaction fractions of 1.2 and 0.4, respectively, and hydrophobic interactions with Trp286 showing an interaction fraction of approximately 1. For molecule 17c, the residues Trp286, Tyr124, Tyr341, Phe295, Arg296, and His447 demonstrated significant interactions, including hydrogen bonds with Tyr124 and His447 reaching fractions up to 0.8, and hydrophobic interactions with Trp286, and Tyr341 exceeding 0.8 (Fig 10). These results demonstrate the stability of these molecules and support the docking findings, suggesting that both molecules could be promising candidates for AChE inhibition.

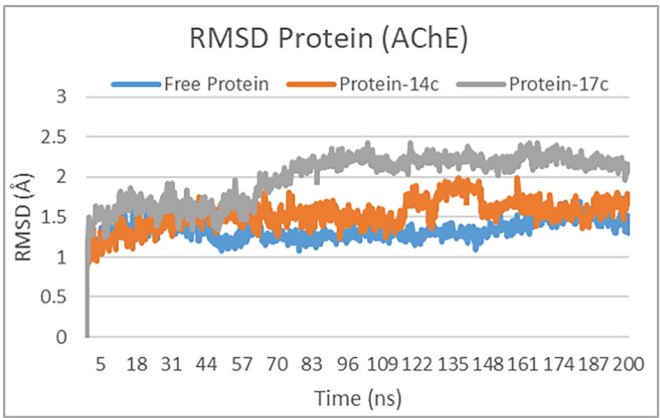
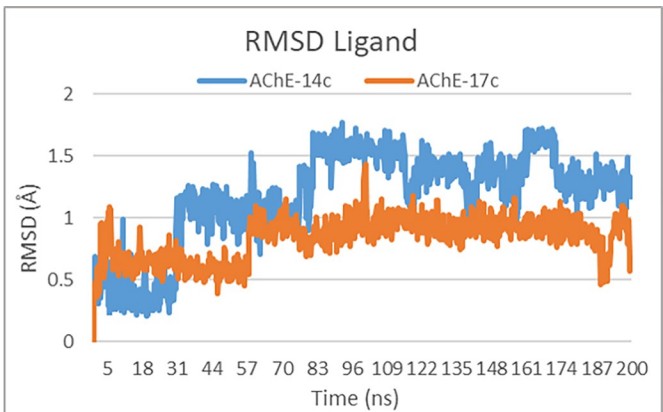

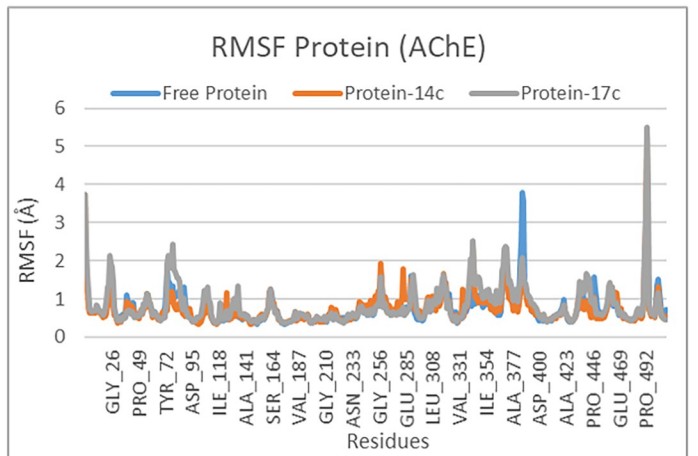

**Fig 9. The RMSD protein, RMSD ligand and RMSF protein plots of both complexes AChE-14c and AChE-17c.**

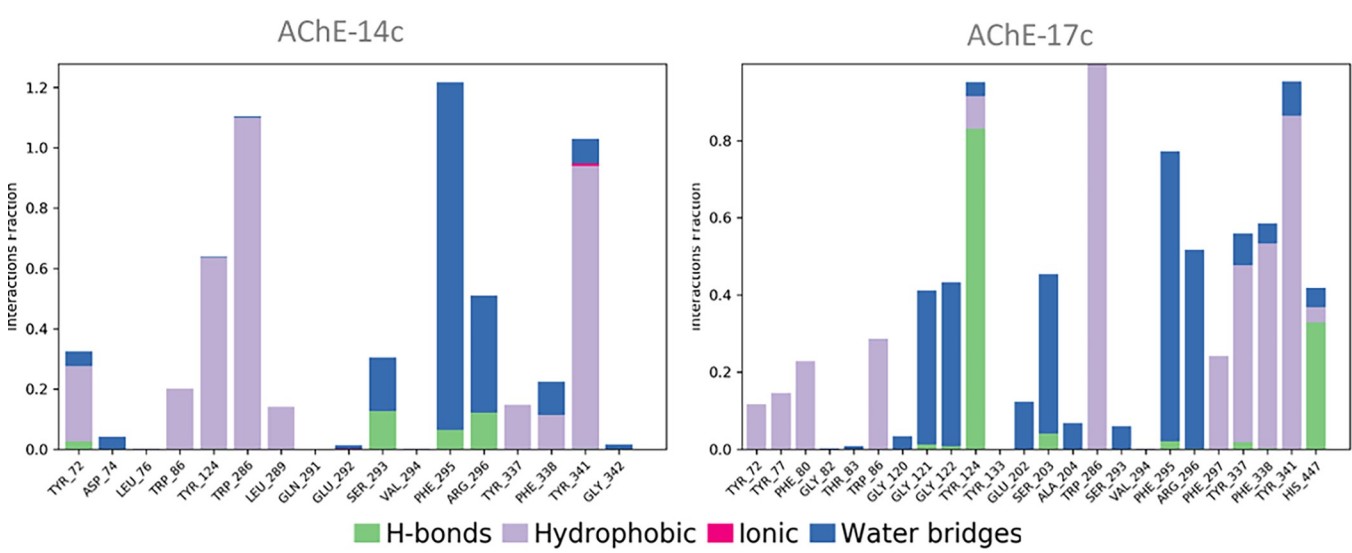

**Fig 10. Protein–ligand interactions of both complexes AChE-14c and AChE-17c.**

For the BuChE target, the RMSD analysis of the protein and its complexes with ligands 14c and 17c shows that all systems reach equilibrium after approximately 10 ns of simulation. The RMSD of the free protein stabilizes at an average deviation of about 1.5 Å, reflecting a relatively rigid and stable structure. Similarly, the protein-ligand complex with 14c shows a stabilized RMSD around 1.5 Å, suggesting only minor conformational changes. However, the complex with ligand 17c exhibits a slightly higher RMSD, reaching approximately 2 Å, which implies that this ligand induces more significant conformational modifications while maintaining overall stability (Fig 11). Regarding the RMSD analysis of the ligands, ligand 14c displays minimal fluctuations, oscillating around 1.5 Å before stabilizing around 2 Å towards the end of the simulation. Meanwhile, the RMSD of ligand 17c indicates stability around 1.5 Å, accompanied by slightly greater fluctuations after 100 ns (Fig 11).

The RMSF analysis of the protein residues reveals that most residues in BuChE show low flexibility in both the free protein and its complexes, particularly in the presence of ligand 14c. However, certain key residues, such as Gln380, Ala328, and Fpk710, exhibit increased flexibility, especially in the BuChE-17c complex (Fig 11). These results suggest that both ligands 14c and 17c maintain stable interactions with the BuChE protein, while ligand 17c leads to more pronounced conformational adjustments.

The protein-ligand interaction analysis for molecules 14c and 17c reveals notable differences in their binding modes with the BuChE protein. Molecule 14c exhibits dominant

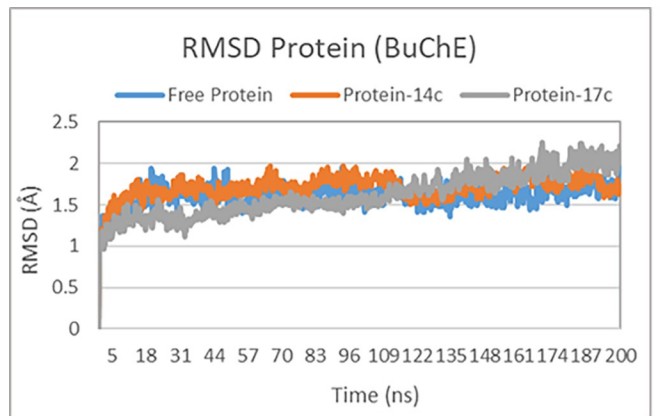 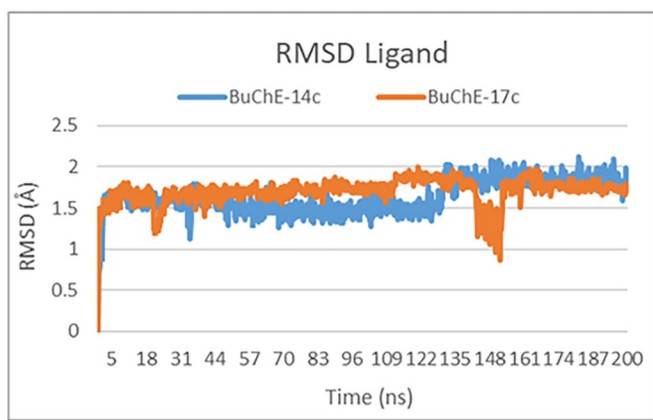

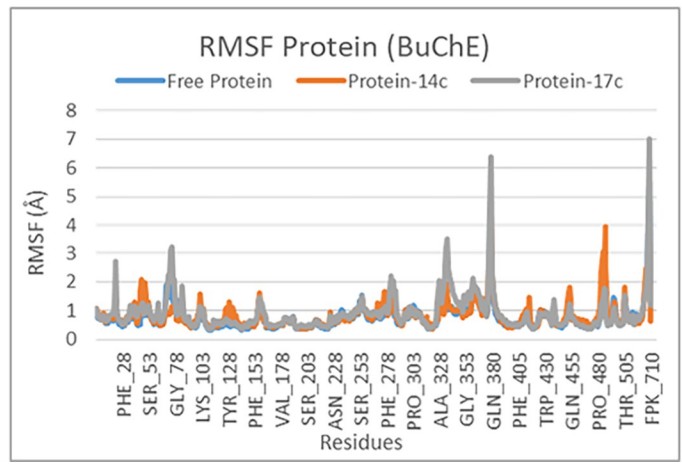

**Fig 11. The RMSD protein, RMSD ligand and RMSF protein plots of both complexes BuChE-14c and BuChE-17c.**

interactions with the residues Trp82, Gly115, and Thr120, primarily through hydrophobic interactions and hydrogen bonds, with interaction fractions of 1.6, 0.8, and 0.8, respectively. Molecule 17c shows very strong interactions with Trp82 and Tyr128, with interaction fractions of approximately 1.2 and 1.0, respectively, and significant interactions with Thr120, around 0.4. Additionally, residues Tyr332, Gly115, and Phe329 also contribute, albeit to a lesser extent. Molecule 17c displays a broader diversity of interactions, including hydrophobic, hydrogen, and ionic interactions. This variety accounts for the larger conformational fluctuations observed in the RMSD analysis of the BuChE-17c complex (Fig 12). Consequently, although both ligands interact effectively with BuChE, ligand 17c induces more pronounced conformational adjustments, suggesting more dynamic and potentially more stable interactions with the protein.

For the MAO-B target, the RMSD analysis reveals that the free protein starts with a deviation of about 1 Å at the beginning of the simulation, which gradually increases to a value between 1.5 and 2 Å after 200 ns, indicating relative stability throughout the simulation. The protein-ligand complexes show slightly different behavior. The RMSD for the protein-14c complex exhibits more pronounced fluctuations, reaching up to 2.5 Å between 50 and 100 ns, before stabilizing around 2 Å for the remainder of the simulation. On the other hand, the protein-17c complex shows more moderate fluctuations, with the RMSD stabilizing between 1.5 and 2 Å after an adaptation phase (Fig 13). Regarding the ligands, the RMSD analysis shows that ligand 14c undergoes more significant fluctuations, reaching up to 1.5 Å throughout the 200 ns of simulation, while ligand 17c demonstrates greater stability, with RMSD deviations fluctuating around 1 Å. These results suggest that the interaction of ligand 17c with the protein is more stable and consistent than that of ligand 14c (Fig 13).

The RMSF analysis shows that the free protein exhibits moderate fluctuations, indicating increased rigidity in certain regions. In the presence of ligand 14c, the residue fluctuations remain generally similar to those of the free protein. However, marked peaks are observed at residues Pro458 and Leu482, reaching 4 Å and 8 Å, respectively, suggesting increased flexibility in these specific areas. For the complex with ligand 17c, the fluctuations remain comparable to those seen with the free protein, with no significant changes in overall flexibility (Fig 13).

The protein-ligand interactions Analysis reveals that molecule 14c forms several interactions, with the most dominant observed with the Tyr188 residue, having an interaction

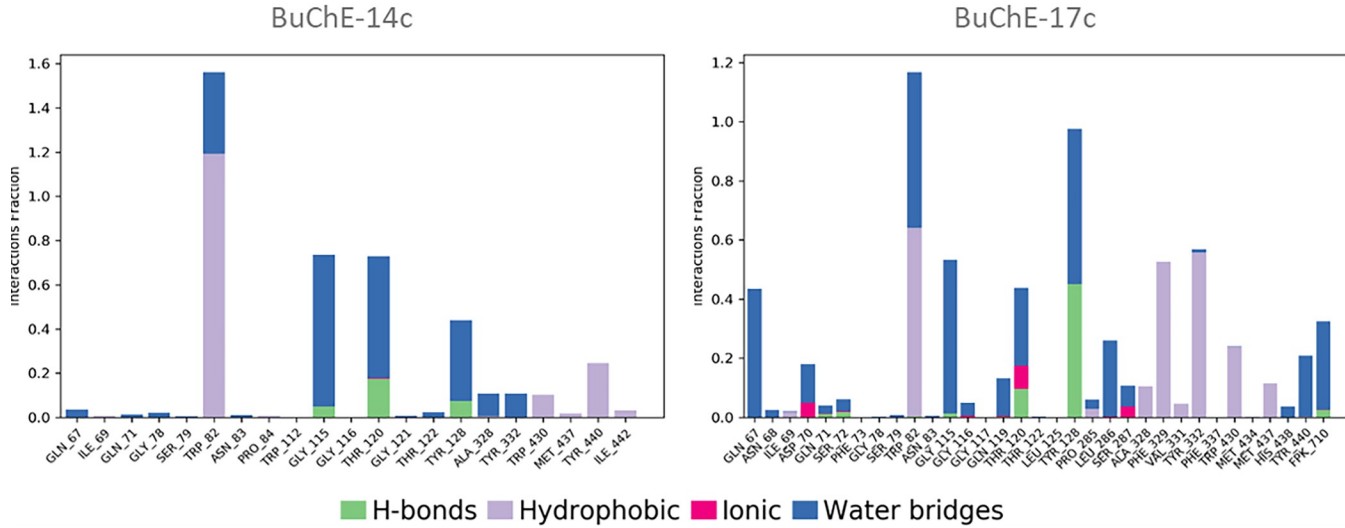

**Fig 12. Protein–ligand interactions of both complexes BuChE-14c and BuChE-17c.**

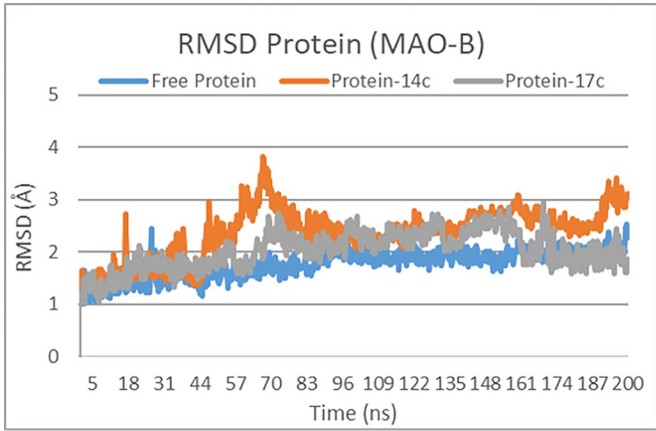

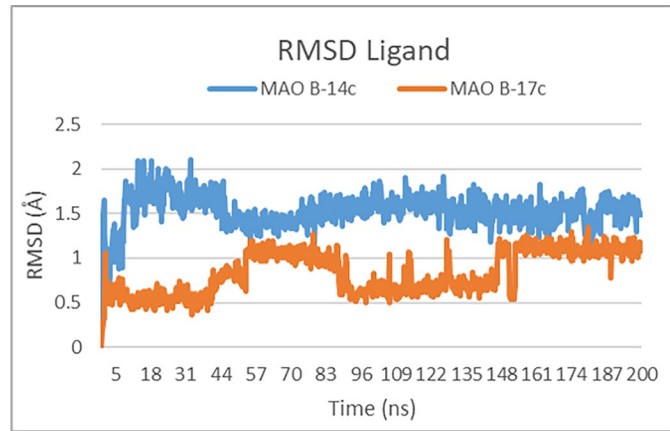

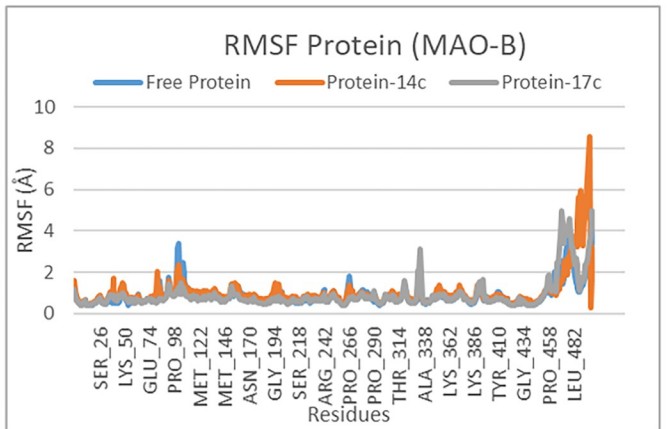

**Fig 13. The RMSD protein, RMSD ligand and RMSF protein plots of both complexes MAO-B-14c and MAO-B-17c.**

fraction reaching 1.0, followed by significant contributions from Tyr435 (approximately 0.7) and Cys172 (approximately 0.6). These interactions are primarily hydrophobic, accompanied by hydrogen bonds, suggesting a strong and stable binding pattern. For molecule 17c, the most dominant interactions include bindings with Tyr398, Tyr60, and Gln206, with binding fractions exceeding 0.6, primarily consisting of hydrophobic interactions and hydrogen bonds (Fig 14). This broader distribution of interactions reflects the diverse ways these two ligands modulate the activity of the MAO-B enzyme.

## MM-GBSA calculations

MM-GBSA calculation was used to evaluate the binding energy (ΔG_bind) of complexes formed by compounds 14c and 17c with three protein targets: AChE, BuChE, and MAO-B. This method provides an estimate of the thermodynamic stability of the complexes by integrating the contributions of molecular mechanics force fields and solvation effects [51]. The ΔG_bind values, presented in Table 5, reflect the strength of the interactions between the ligands and their protein targets. A more negative ΔG_bind value indicates a stronger binding and a more favourable interaction, suggesting greater complex stability. The results show that molecule 14c exhibits particularly favourable interactions with AChE (ΔG_bind of -73.32 kcal/mol), BuChE (ΔG_bind of -68.72 kcal/mol), and MAO-B (ΔG_bind of -63.01 kcal/mol), indicating strong affinity and stable interactions with these targets. Molecule 17c, on the other

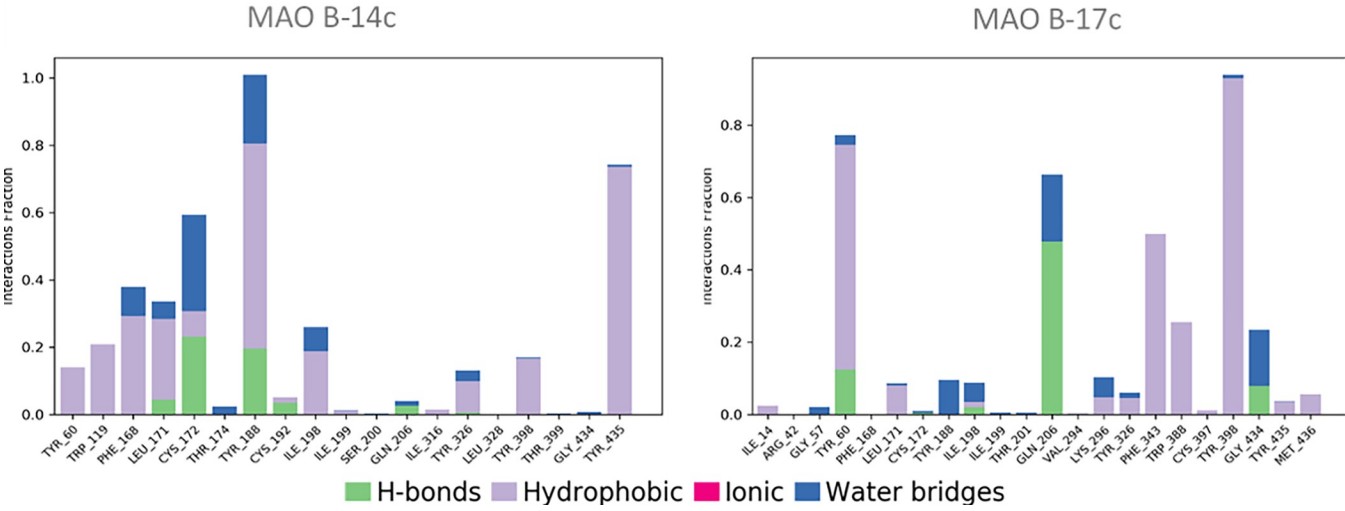

**Fig 14. Protein–ligand interactions of both complexes MAO-B-14c and MAO-B-17c.**

hand, shows an even stronger interaction with AChE (ΔG_bind of -79.86 kcal/mol) and has a significative affinity for BuChE (ΔG_bind of -53.27 kcal/mol) compared to ligand 14c, and presents a ΔG_bind of -65.07 kcal/mol for MAO B, which is slightly more favourable than that of 14c. These results indicate that molecule 14c is particularly effective for BuChE and well-suited for AChE and MAO-B, while molecule 17c is particularly effective for AChE, favourable for MAO B, but less effective for BuChE. Both ligands, 14c and 17c, show relative stability, highlighting their potential as multi-target inhibitors in the pathology of AD.

## Conclusion

In this study, a series of novel molecules derived from 8-hydroxyquinoline (Azo-8HQ) was proposed and designed due to their promising pharmacological properties for treating AD. We focused on three enzymes implicated in AD pathology: AChE, BuChE, and MAO-B. A comprehensive in silico study was conducted, including docking studies to evaluate the binding affinity of the molecules and ADMET profiling to estimate their pharmacokinetic properties and safety. Additionally, DFT studies were performed to gain deeper insights into the electronic properties of the molecules. Molecular dynamics simulations followed by MM-GBSA calculations were carried out to reliably examine the binding of these molecules to the enzymes and assess their stability. The results showed that, among the 63 derivatives in the series, two molecules, 14c and 17c, demonstrated strong affinities for the three studied targets, as evidenced by the interactions formed by these complexes compared to reference compounds. These two molecules also revealed a favourable pharmacokinetic profile. Furthermore, DFT studies highlighted favourable electronic properties, and the molecular dynamics simulations demonstrated the stability of the molecules in the active sites of the targets over a

**Table 5. MM-GBSA binding affinity results for the selected compounds.**

| Molecule | MM-GBSA Binding Affinities (Kcal/mol) | | |
|---|---|---|---|
| | **AChE** | **BuChE** | **MAO-B** |
| **14c** | -73.32 | -68.72 | -63.01 |
| **17c** | -79.86 | -53.27 | -65.07 |

200 ns period under physiological conditions. The MM-GBSA calculations for the complexes formed by molecules 14c and 17c with the studied targets showed low binding energies, indicating notable stability of these molecules. Consequently, these two molecules were identified in silico as potential multi-target candidates against the enzymes AChE, BuChE, and MAO-B, primarily involved in AD pathology, based on an in-depth in silico study aimed at minimizing drug development costs and time. It is worth noting that despite all of these positive results, from a medicinal chemistry viewpoint the molecules do present a few potential metabolic liabilities which will need to considered more closely upon synthesis, and further evaluation (e.g., terminal alkynes, diazenyl motifs). As for future outlook, this research will progress with further investigations, including the synthesis of these two lead compounds and detailed *in vitro* and *in vivo* testing to assess their potential medicinal use against AD.

## Supporting information

**S1 Table. 2D structures of the Azo-8HQ molecules studied.**
(DOCX)

## Author Contributions

**Conceptualization:** Fatima Zahra Guerguer, Bouchra Rossafi, Oussama Abchir.

**Data curation:** Fatima Zahra Guerguer, Oussama Abchir, Dhabya Bakhit Albalushi.

**Formal analysis:** Fatima Zahra Guerguer.

**Funding acquisition:** Yasir S. Raouf, Abdelouahid Samadi.

**Investigation:** Samir Chtita.

**Methodology:** Samir Chtita.

**Project administration:** Samir Chtita.

**Resources:** Samir Chtita.

**Software:** Samir Chtita.

**Supervision:** Samir Chtita.

**Validation:** Yasir S. Raouf, Abdelouahid Samadi, Samir Chtita.

**Visualization:** Bouchra Rossafi, Yasir S. Raouf, Abdelouahid Samadi.

**Writing – original draft:** Fatima Zahra Guerguer, Bouchra Rossafi, Oussama Abchir.

**Writing – review & editing:** Fatima Zahra Guerguer, Bouchra Rossafi, Oussama Abchir, Dhabya Bakhit Albalushi.

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
