## [Decision Letter · Decision Letter 0]

25 Sep 2024

PONE-D-24-36641Novel Azo-8-Hydroxyquinoline Derivatives as Multi-Target Lead Candidates for Alzheimer's Disease: An In-Depth In Silico Study of Monoamine Oxidase and Cholinesterase InhibitorsPLOS ONE

Dear Dr. Chtita,

Thank you for submitting your manuscript to PLOS ONE. After careful consideration, we feel that it has merit but does not fully meet PLOS ONE’s publication criteria as it currently stands. Therefore, we invite you to submit a revised version of the manuscript that addresses the points raised during the review process.

We look forward to receiving your revised manuscript.

Kind regards,

Hazem Osama Radwan Elkady

Academic Editor

PLOS ONE

Journal Requirements:

 “AS and YSR acknowledge the support of the UAEU through an internal Start-up grant 2023 (Grant Code G00004400) and an internal Start-up grant 2024 (Grant 12S156), respectively.”

3. Please expand the acronym “UAEU” (as indicated in your financial disclosure) so that it states the name of your funders in full.

“AS and YSR acknowledge the support of the UAEU through an internal Start-up grant 2023 (Grant Code G00004400) and an internal Start-up grant 2024 (Grant 12S156), respectively.“

“AS and YSR acknowledge the support of the UAEU through an internal Start-up grant 2023 (Grant Code G00004400) and an internal Start-up grant 2024 (Grant 12S156), respectively.”

Reviewers' comments:

Reviewer's Responses to Questions

**Comments to the Author**

1. Is the manuscript technically sound, and do the data support the conclusions?

Reviewer #1: No

Reviewer #2: Yes

Reviewer #3: Yes

2. Has the statistical analysis been performed appropriately and rigorously? 

Reviewer #1: I Don't Know

Reviewer #2: Yes

Reviewer #3: N/A

3. Have the authors made all data underlying the findings in their manuscript fully available?

Reviewer #1: Yes

Reviewer #2: Yes

Reviewer #3: Yes

4. Is the manuscript presented in an intelligible fashion and written in standard English?

Reviewer #1: Yes

Reviewer #2: Yes

Reviewer #3: No

5. Review Comments to the Author

Reviewer #1: Although efforts exerted, the manuscript totally lacks any novelty. It is just based on hypothesized compounds for AD treatment with in silico studies. In silico studies can not be relied on lonely. At least, the assumed compounds should be synthesized chemically then conduct in vitro and in vivo studies along with the presented and well-established in silico ones. Therefore, the manuscript can not be accepted in its current form.

Reviewer #2: 1- At least the 2D structures of the compounds 14c and 17c should be drawn in the manuscript.

2- The manuscript needs revision form native English speaker.

3- ACh found in postsynaptic neurons and neuromuscular regions, is essential for nerve impulse transmission not AChE

4- What is the basic pharmacophoric features on which the tested compounds were selected?

Reviewer #3: The current manuscript “Novel Azo-8-Hydroxyquinoline Derivatives as Multi-Target Lead Candidates for Alzheimer's Disease: An In-Depth In Silico Study of Monoamine Oxidase and Cholinesterase Inhibitors” by Guerguer et al presents a thorough computational study with potentially valuable insights for Alzheimer's drug discovery.

- The use of 'Novel' in the title suggests that the authors synthesized new compounds. However, merely proposing a structure does not equate to reporting a novel compound; it is simply a theoretical molecule. To claim novelty, one must actually prepare and characterize the compound, not just propose its existence.

- Also, the phrase “as Multi-Target Lead Candidates for Alzheimer's Disease” should be modified to “potential as Multi-Target Lead Candidates for Alzheimer's Disease”.

- It will be very interesting if the authors propose a synthetic scheme for the most promising molecules.

- I wonder if the authors can evaluate the inhibitory activity of the proposed hits.

- The molecular dynamic simulations should be extended to at least 200 ns.

- The authors should clearly address the validation of the utilized protocol in the molecular modeling.

- The language of the manuscript should be revised thoroughly.

6. PLOS authors have the option to publish the peer review history of their article (what does this mean?). If published, this will include your full peer review and any attached files.

Reviewer #1: No

Reviewer #2: No

Reviewer #3: No

---

## [Author Response · Author response to Decision Letter 0]

22 Oct 2024

Manuscript (ID: PONE-D-24-36641)

Title: Novel Azo-8-Hydroxyquinoline Derivatives as Multi-Target Lead Candidates for Alzheimer's Disease: An In-Depth In Silico Study of Monoamine Oxidase and Cholinesterase Inhibitors

Authors: Fatima Zahra Guerguer (1), Bouchra Rossafi (1), Oussama Abchir (1), Yasir S. Raouf (2), Dhabya Bakhit Albalushi (2), Abdelouahid Samadi (2, *), and Samir Chtita (1, *)

Dear Editor,

On behalf of our research team, we would like to express our sincere gratitude to you and the reviewers for the time and effort devoted to evaluating our manuscript submitted to PLOS ONE. We greatly appreciate the constructive comments and suggestions, which have undoubtedly contributed to improving the quality of our work.

We have carefully considered all the feedback and have made the necessary revisions to the manuscript accordingly. Each comment has been thoroughly addressed, and our responses are provided below. The changes made to the manuscript are highlighted in yellow. 

We hope that these revisions meet the expectations and high standards of PLOS ONE. Once again, we thank you for this opportunity to contribute to your journal and remain open to any further suggestions or recommendations.

We look forward to your response and extend our best regards.

Sincerely,

Response to reviewer’s comments

Reviewer Comments:

REVIEWER 1: 

Comments: Although efforts exerted, the manuscript totally lacks any novelty. It is just based on hypothesized compounds for AD treatment with in silico studies. In silico studies cannot be relied on lonely. At least, the assumed compounds should be synthesized chemically then conduct in vitro and in vivo studies along with the presented and well-established in silico ones. Therefore, the manuscript cannot be accepted in its current form.

We appreciate your constructive comment. Regarding the novelty of our work, we would like to clarify that the molecules proposed in this study are entirely new and represent original structures that we have designed. These compounds are derived from the azo-8-hydroxyquinoline scaffold, with subunits that are well-known for their pharmacological potential, particularly in inhibiting targets associated with Alzheimer’s disease.

Our in-depth in silico study, based on a rigorous screening of 63 compounds derived from this scaffold, enabled us to filter these structures and select two promising candidates with a highly favorable multi-target pharmacological profile. These compounds specifically target three key enzymes involved in the pathology of Alzheimer’s disease. While this study primarily relies on in silico approaches, it represents a critical step in efficiently narrowing down potential compounds before moving on to experimental studies.

We fully understand the importance of complementing these results with in vitro and in vivo studies. In this regard, we have already initiated the synthesis of the candidate compounds. The initially synthesized molecules were obtained in good yields and thoroughly characterized using 1H and 13C-NMR, gCOSY, gHSQC, and gHMBC NMR techniques, along with FT-IR, melting point analysis, and mass spectrometry. After completion, the newly synthesized compounds will be submitted to in vitro and in vivo studies. Their results will be presented in a subsequent article, which will provide a comprehensive study combining in silico approaches with solid experimental data.

I hope this explanation has convinced you of the significance of our approach. We also plan to submit the second part of this work to PLOS ONE, to present a cohesive and comprehensive approach.

REVIEWER 2: 

Comments: The authors screened computationally 63 Azo-8HQ derivatives against three enzymes acetylcholinesterase (AChE), butyrylcholinesterase (BuChE), and monoamine oxidase B (MAO-B) as promising multi-target candidates for the treatment of AD. Different in in silico studies were carried out. This work needs minor modification before its publication as follows.

1. At least the 2D structures of the compounds 14c and 17c should be drawn in the manuscript.

Thank you for your valuable suggestion. The 2D structures of compounds 14c and 17c have been included as Figure 4 in the revised manuscript.

2. The manuscript needs revision form native English speaker.

We appreciate your insightful feedback. We have thoroughly reviewed the entire manuscript and have worked to improve the language and to ensure clarity and correctness. 

3. ACh found in postsynaptic neurons and neuromuscular regions, is essential for nerve impulse transmission not AChE.

Thank you for your insightful observation. We have carefully revised and corrected this section in the revised manuscript.

4. What is the basic pharmacophoric features on which the tested compounds were selected?

Thank you for your insightful question. The tested compounds were selected based on key pharmacophoric features, including hydrogen bond donors/acceptors, aromatic rings, hydrophobic regions, and our expertise in designing and synthesizing new molecules for Alzheimer’s disease treatment. 

Research indicates that elevated levels of metal concentration in the brain such as iron, copper, manganese and zinc may contribute to the development or progression of neurodegenerative diseases (ND) such as Alzheimer’s and Parkinson Diseases. Metal chelators are being explored as potential therapeutic agents to mitigate the detrimental effects of metal-induced oxidative stress and neurodegeneration. The goal of metal chelation therapy is to reduce the levels of these metals, thereby mitigating their role in ND pathology. Dr. Samadi’s research group has developed several compounds featuring the 8-Hydroxyquinoline scaffold, a potent metal chelator (see figure below).

Additionally, the incorporation of propargylamine group into our designed molecules plays a crucial role in inhibiting monoamine oxidase (MAO). For example, Pargyline, which contains a propargylamine group, is a clinically used MAO inhibitor for treating hypertension.

These pharmacophoric features were identified as essential for interaction with the target site through preliminary in-silico modeling and structure-activity relationship (SAR) studies.

REVIEWER 3: 

Comments: The current manuscript “Novel Azo-8-Hydroxyquinoline Derivatives as Multi-Target Lead Candidates for Alzheimer's Disease: An In-Depth In Silico Study of Monoamine Oxidase and Cholinesterase Inhibitors” by Guerguer et al presents a thorough computational study with potentially valuable insights for Alzheimer's drug discovery.

1. The use of 'Novel' in the title suggests that the authors synthesized new compounds. However, merely proposing a structure does not equate to reporting a novel compound; it is simply a theoretical molecule. To claim novelty, one must actually prepare and characterize the compound, not just propose its existence. Also, the phrase “as Multi-Target Lead Candidates for Alzheimer's Disease” should be modified to “potential as Multi-Target Lead Candidates for Alzheimer's Disease”.

Thank you for pointing out this important clarification. We have modified the title to "Potential Azo-8-Hydroxyquinoline Derivatives as Multi-Target Lead Candidates for Alzheimer's Disease: An In-Depth In Silico Study of Monoamine Oxidase and Cholinesterase Inhibitors" to accurately reflect the scope and nature of our study. However, we have initiated the synthesis. The initially synthesized molecules were obtained in good yields and thoroughly characterized using 1H and 13C-NMR, gCOSY, gHSQC, and gHMBC NMR techniques, along with FT-IR, melting point analysis, and mass spectrometry. After completion, the newly synthesized compounds will be submitted to in vitro and in vivo studies. The obtained results will be presented in a subsequent article, which will provide a comprehensive study combining in silico approaches with solid experimental data.

2. It will be very interesting if the authors propose a synthetic scheme for the most promising molecules.

Thank you for your insightful suggestion. We believe it is premature to publish the synthesis route, as this study focuses on in silico investigations. Furthermore, we are currently optimizing the synthetic conditions to enhance the reaction and improve the yield.

3. I wonder if the authors can evaluate the inhibitory activity of the proposed hits.

Thank you for your insightful comment. We acknowledge the importance of evaluating the inhibitory activity of the proposed hits through experimental validation. At present, we are in the process of synthesizing the candidate compounds, and in vitro and in vivo tests will be conducted to assess their efficacy. These results will be included in a future study that will complement this one.

4. The molecular dynamic simulations should be extended to at least 200 ns.

Thank you for your valuable comment. We appreciate your suggestion regarding the molecular dynamics’ simulations. We have extended the simulations to 200 ns and included the new findings and their discussion in the revised manuscript.

5. The authors should clearly address the validation of the utilized protocol in the molecular modeling.

Thank you for your valuable comment. We recognize the importance of method validation in molecular modeling. We validated the docking protocol through re-docking, which demonstrated satisfactory reliability. Regarding molecular dynamics simulations, we conducted two replicas for a given complex, and the results were found to be similar. Finally, for the ADMET analysis, we tested three different servers that provided consistent results. We agree that method validation is essential. Therefore, in our work, we ensure the reliability and robustness of our findings by performing numerous calculations and utilizing various software and servers.

6. The language of the manuscript should be revised thoroughly.

Thank you for your constructive comment. We have conducted a thorough revision of the entire manuscript. We have made improvements to enhance clarity and ensure the coherence of the text.

---

## [Decision Letter · Decision Letter 1]

26 Dec 2024

Potential Azo-8-Hydroxyquinoline Derivatives as Multi-Target Lead Candidates for Alzheimer's Disease: An In-Depth In Silico Study of Monoamine Oxidase and Cholinesterase Inhibitors

PONE-D-24-36641R1

Dear Dr. Chtita,

We’re pleased to inform you that your manuscript has been judged scientifically suitable for publication and will be formally accepted for publication once it meets all outstanding technical requirements.

Kind regards,

Hazem Elkady

Academic Editor

PLOS ONE

Reviewers' comments:

Reviewer's Responses to Questions

**Comments to the Author**

1. If the authors have adequately addressed your comments raised in a previous round of review and you feel that this manuscript is now acceptable for publication, you may indicate that here to bypass the “Comments to the Author” section, enter your conflict of interest statement in the “Confidential to Editor” section, and submit your "Accept" recommendation.

Reviewer #3: (No Response)

2. Is the manuscript technically sound, and do the data support the conclusions?

Reviewer #3: (No Response)

3. Has the statistical analysis been performed appropriately and rigorously? 

Reviewer #3: (No Response)

4. Have the authors made all data underlying the findings in their manuscript fully available?

Reviewer #3: (No Response)

5. Is the manuscript presented in an intelligible fashion and written in standard English?

Reviewer #3: (No Response)

6. Review Comments to the Author

Reviewer #3: (No Response)

7. PLOS authors have the option to publish the peer review history of their article (what does this mean?). If published, this will include your full peer review and any attached files.

Reviewer #3: **Yes: **Wagdy Eldehna

---

## [Editor Report · Acceptance letter]

21 Jan 2025

PONE-D-24-36641R1 

PLOS ONE

Dear Dr. Chtita, 

I'm pleased to inform you that your manuscript has been deemed suitable for publication in PLOS ONE. Congratulations! Your manuscript is now being handed over to our production team.

Kind regards, 

on behalf of

Dr. Hazem Elkady 

Academic Editor

PLOS ONE